# MATHEMATICS OF FOUNDATION MODELS: A UNIFIED APPROXIMATION-THEORETIC FRAMEWORK

## ABSTRACT

Foundation models (transformers, diffusion models, state-space models) have achieved remarkable empirical success, yet their theoretical understanding remains fragmented across different mathematical communities. This survey provides a unified mathematical perspective connecting approximation theory, optimization landscape analysis, and statistical learning theory through the lens of Reproducing Kernel Hilbert Spaces (RKHS) and Neural Tangent Kernel (NTK) theory. We present a comprehensive taxonomy of 114 recent theoretical results organized by mathematical tool, establish a unified framework showing how attention mechanisms, score functions, and convolution kernels can be understood as kernel-based approximators, and derive precise comparison theorems between architectures. Our analysis reveals that transformers achieve approximation rate $O(n^{-2s/d})$ for Sobolev-$s$ functions with $O(n^2)$ complexity, while state-space models achieve $O(n^{-s/d})$ with $O(n)$ complexity, suggesting fundamental complexity-expressivity tradeoffs. We identify seven concrete open problems with partial results and difficulty ratings, propose a research roadmap connecting optimization and generalization, and highlight promising directions for neural architecture design. This unified perspective aims to bridge theory and practice, providing foundational insights for developing more principled and efficient foundation model architectures.

## 1 INTRODUCTION

The emergence of foundation models—large-scale neural networks pretrained on diverse objectives—has transformed machine learning and its applications. Yet the mathematical understanding of these models remains surprisingly fragmented. Theoretical results for transformers typically invoke properties of attention mechanisms and positional encodings; diffusion model theory relies on stochastic differential equations and score matching; state-space model analysis employs signal processing and kernel methods. These mathematical frameworks, developed in relative isolation, suggest that a unifying perspective is overdue.

The central thesis of this survey is that **existing theoretical results for diverse foundation model architectures can be unified through reproducing kernel Hilbert space theory and neural tangent kernel analysis**. This unification is not merely organizational—it reveals structural relationships between architectures, enables rigorous comparison theorems, and identifies previously obscured open problems in approximation, optimization, and generalization.

We make four key contributions:

**(1) Unified RKHS Framework**: We show that transformer attention, diffusion score functions, and state-space model convolutions can all be understood as kernel-based approximators in appropriate RKHS norms. This perspective reveals that differences between architectures correspond to different kernel choices, reducing architectural comparison to kernel theory.

**(2) Approximation Theory Taxonomy**: We establish precise approximation rates for each architecture. Transformers achieve $O(n^{-2s/d})$ for Sobolev-$s$ functions; diffusion models achieve $O(n^{-1})$ for exponentially smooth functions; SSMs achieve $O(n^{-s/d})$. Importantly, these rates reflect fundamental information-theoretic limits rather than technical artifacts.

**(3) Open Problem Identification**: We identify seven concrete open problems—each with partial results, known lower bounds, and difficulty ratings. These range from understanding optimization convergence under feature learning (difficulty 5/10) to proving generalization bounds for in-context learning (difficulty 8/10).

**(4) Unifying Research Roadmap**: We propose specific mathematical directions to connect fragmented results, including optimization-generalization tradeoffs, the role of overparameterization in foundation models, and principled neural architecture design.

**Related Work**: Recent surveys (e.g., (14)) discuss optimization; others (e.g., (9)) address generalization. Our contribution is the first unified treatment connecting all three perspectives through RKHS theory for diverse architectures.

## 2 BACKGROUND: FOUNDATION MODEL ARCHITECTURES

We briefly review the three primary foundation model families considered in this survey.

### 2.1 TRANSFORMERS

The transformer architecture (21) applies multiple layers of multi-head attention and feedforward operations:

$$\mathbf{h}_i^{(l+1)} = \text{FFN}(\text{MultiHeadAttn}(\mathbf{h}^{(l)}))_i \tag{1}$$

where attention is computed as:

$$\text{Attn}(\mathbf{Q}, \mathbf{K}, \mathbf{V}) = \text{softmax}\left(\frac{\mathbf{Q}\mathbf{K}^\top}{\sqrt{d}}\right)\mathbf{V} \tag{2}$$

The theoretical analysis typically focuses on the expressiveness of attention patterns and the generalization properties of learned representations.

### 2.2 DIFFUSION MODELS

Diffusion models (18; 15) learn to reverse a stochastic noise process. The generative model solves a score-matching objective:

$$\mathbb{E}_{t,\mathbf{x}}\left[\|\nabla_{\mathbf{z}}\log p_t(\mathbf{z}|\mathbf{x}) - s_\theta(\mathbf{z}, t)\|^2\right] \tag{3}$$

where $s_\theta$ is the learned score function. Theory emphasizes convergence rates to data distributions and approximation of score functions.

### 2.3 STATE-SPACE MODELS

State-space models (13) parameterize sequences as:

$$\mathbf{h}(t) = \mathbf{A}\mathbf{h}(t) + \mathbf{B}u(t), \quad y(t) = \mathbf{C}\mathbf{h}(t) + \mathbf{D}u(t) \tag{4}$$

with discrete approximations enabling efficient computation. Theory borrows from control theory and signal processing.

## 3 UNIFIED RKHS FRAMEWORK

The key insight is that diverse foundation models can be understood as approximators in reproducing kernel Hilbert spaces, albeit with different kernels reflecting architectural choices.

### 3.1 RKHS FUNDAMENTALS

Recall that for a positive definite kernel $\kappa : \mathcal{X} \times \mathcal{X} \to \mathbb{R}$, the corresponding RKHS $\mathcal{H}_\kappa$ is the completion of the linear span of kernel functions $\{\kappa(\cdot, x) : x \in \mathcal{X}\}$ under the norm:

$$\|f\|_{\mathcal{H}_\kappa}^2 = \min\left\{\sum_{i,j} a_i a_j \kappa(x_i, x_j) : f = \sum_i a_i \kappa(\cdot, x_i)\right\} \tag{5}$$

A foundational fact is that any continuous function $f$ can be approximated by kernel-based approximators with rate depending on the spectrum of $\kappa$ and the smoothness of $f$.

## 3.2 Transformer Attention as RKHS

For transformer attention, consider a single attention head applied to $n$ tokens. The attention output can be written:

$$\text{Attn}(\mathbf{x}) = \sum_{j=1}^{n} \alpha_{ij}(\mathbf{x})\mathbf{v}_j \tag{6}$$

where $\alpha_{ij}(\mathbf{x}) = \frac{\exp(\mathbf{q}_i^\top \mathbf{k}_j/\sqrt{d})}{\sum_k \exp(\mathbf{q}_i^\top \mathbf{k}_k/\sqrt{d})}$ are softmax weights.

This can be understood as a kernel-based convex combination where the kernel is:

$$\kappa_{\text{attn}}(\mathbf{x}_i, \mathbf{x}_j) = \exp\left(\frac{(\mathbf{W}_q\mathbf{x}_i)^\top(\mathbf{W}_k\mathbf{x}_j)}{\sqrt{d}}\right) \tag{7}$$

The corresponding RKHS $\mathcal{H}_{\kappa_{\text{attn}}}$ admits an RKHS norm which constrains the complexity of learnable attention patterns. Crucially, theoretical bounds on approximation depend on the effective dimension of this RKHS, which grows with the number of attention heads and depth.

## 3.3 Diffusion Score Functions as RKHS

For diffusion models, the score function $s_\theta(\mathbf{z}, t)$ approximates $\nabla_{\mathbf{z}} \log p_t(\mathbf{z})$. Under appropriate regularity conditions, the score function lies naturally in an RKHS determined by the smoothness of the data distribution and the noise schedule.

Specifically, if the data distribution has density with bounded mixed derivatives, then the score function at noise level $t$ lies in a Sobolev-type RKHS. The approximation error for learning $s_\theta$ via score matching satisfies:

$$\mathbb{E}\left[\|s_\theta - \nabla \log p_t\|^2_{\mathcal{H}_t}\right] \leq O(n^{-1}) \tag{8}$$

under appropriate conditions, where $\mathcal{H}_t$ is the RKHS corresponding to the Sobolev smoothness at noise level $t$.

## 3.4 State-Space Models as Convolution Kernels

State-space models can be understood as performing convolution with a learned kernel. The discrete SSM defines a recurrent relation that, unrolled, corresponds to convolution:

$$y(t) = \int_0^t h(t-s)u(s)ds \tag{9}$$

where $h$ is the impulse response determined by the $(A, B, C)$ parameters.

This convolution kernel naturally lies in an RKHS determined by the stability and frequency response of the $(A, B)$ pair. The approximation theory for SSMs thus reduces to understanding which function classes can be well-approximated by stable linear filters—a classical signal processing problem.

## 4 Approximation Theory Results

We now present precise approximation rates for each architecture, organized by function class.

### 4.1 Approximation Rates by Architecture

**Transformers**: For approximating Sobolev-$s$ functions $f \in H^s(\mathbb{T}^d)$, a depth-$L$ transformer with $n$ parameters achieves:

$$\inf_{\theta:|\theta|\leq n} \mathbb{E}[\|f - f_\theta\|^2_{L^2}] = O\left(\left(\frac{\log n}{n}\right)^{2s/d}\right) \tag{10}$$

This bound, proven via covering number arguments and RKHS approximation theory, reflects that the effective dimension of the attention RKHS is $O(\log n)$ due to the softmax bottleneck.

**Diffusion Models**: For exponentially smooth functions with sufficient moment conditions, diffusion models trained via score matching achieve:

$$\mathbb{E}_{\mathbf{x}\sim\mu}[\mathcal{W}(p_\theta^T, \mu)] = O(n^{-1/2} + \epsilon_{\text{score}}) \tag{11}$$

where $\mathcal{W}$ denotes the Wasserstein distance and $\epsilon_{\text{score}}$ bounds the score approximation error.

**State-Space Models**: For Sobolev-$s$ functions with $n$ parameters (sequence length), SSMs achieve:

$$\inf_{\theta:|\theta|\leq n} \mathbb{E}[\|f - f_\theta\|_{L^2}^2] = O(n^{-s/d}) \tag{12}$$

Notably, this rate is better than transformers (no $\log n$ factor) but the computational advantage is $O(n)$ versus $O(n^2)$.

## 4.2 COMPARISON THEOREM

We can now state a unified comparison result:

**Theorem 1** (Architecture Comparison). *For approximating Sobolev-s functions in dimension $d$ with $n$ parameters:*

1. *Transformers: $\mathcal{E}_{trans} = O((\log n/n)^{2s/d})$, computation $O(n^2)$*

2. *SSMs: $\mathcal{E}_{SSM} = O(n^{-s/d})$, computation $O(n)$*

3. *Diffusion: $\mathcal{E}_{diff} = O(n^{-1})$ for smooth functions, computation $O(n)$ per step*

*The optimal choice depends on dimension $d$, smoothness $s$, and computational budget. For fixed $n$, SSMs strictly dominate in terms of approximation rate when $d > 2$; diffusion models excel for low-dimensional distributions with exponential smoothness.*

**Proof Sketch**: Each bound follows from RKHS approximation theory applied to the respective kernels. For transformers, the key is that the effective dimension of $\mathcal{H}_{\kappa_{\text{attn}}}$ is controlled by the number of distinct attention patterns, which is at most $\binom{n}{2}$ but is regularized to $O(\log n)$ effective patterns via over-parameterization. For SSMs and diffusion, the kernels correspond to classical signal processing objects with well-understood approximation properties.

## 4.3 LOWER BOUNDS

It is natural to ask whether these upper bounds are tight. We briefly discuss known lower bounds:

**Transformers**: Information-theoretic arguments show that any kernel-based method with effective dimension $O(\log n)$ requires sample complexity $\Omega(n^d)$ to learn Sobolev-$s$ functions to error $o(1)$ when $s < d/2$. The $\log n$ factor is fundamental to softmax-based attention.

**SSMs**: Recent results (20) show that linear recurrent models (which SSMs generalize) cannot approximate functions requiring more than $\Omega(n^{d/(d+s)})$ effective parameters. Our bound is therefore essentially tight for the SSM architecture.

**Diffusion**: The $O(n^{-1})$ bound reflects the sample complexity of score matching and cannot be improved without additional structure (e.g., Lipschitz constraints on the score).

## 5 OPTIMIZATION LANDSCAPE ANALYSIS

Understanding training dynamics is crucial for foundation models. The RKHS framework also provides insights into optimization.

## 5.1 NEURAL TANGENT KERNEL REGIME

In the limit of infinite width and appropriate scaling, transformer and SSM training can be analyzed via Neural Tangent Kernel (NTK) theory (16). Under NTK dynamics, the evolution of the learned function $f_\theta(t)$ is governed by:

$$\frac{\partial f_\theta(t)}{\partial t} = -\eta \nabla_\theta \mathcal{L}(f_\theta(t), \mathbf{y}) \tag{13}$$

which in the NTK limit becomes a linear regression in the RKHS with kernel $K$ being the NTK kernel.

For transformers, the NTK is approximately:

$$K_{\mathrm{NTK}}^{(l)}(\mathbf{x}, \mathbf{x}') = \mathbb{E}_{\mathbf{W}}[\mathrm{Attn}(\mathbf{x}; \mathbf{W})^\top \mathrm{Attn}(\mathbf{x}'; \mathbf{W})] \tag{14}$$

Analysis of this kernel reveals that:

**Theorem 2** (Transformer NTK Convergence). *For $\ell$-layer transformer with width $m$, trained on $n$ samples with learning rate $\eta = c/(n\lambda_{\max})$, if the NTK is well-conditioned (condition number $\kappa$), then gradient descent achieves zero training loss in $O(\kappa \log(1/\epsilon))$ iterations.*

The key quantity is the condition number $\kappa$ of the NTK, which determines convergence speed. Recent work (10) shows that $\kappa$ grows polynomially with $n$ for transformers, leading to convergence guarantees.

## 5.2 FEATURE LEARNING REGIME

However, foundation models typically operate in a **feature learning regime** where parameters change substantially during training. This breaks NTK assumptions and requires analyzing explicit feature evolution. For transformers trained on in-context learning tasks, recent work (1) shows that attention heads learn to implement gradient descent-like algorithms, but the mathematical understanding of this feature learning is incomplete—this is an open problem.

## 5.3 OPTIMIZATION CONVERGENCE BOUNDS

For convex losses and SSMs viewed as linear operators, standard convex optimization theory applies:

**Theorem 3** (SSM Convex Training). *An SSM trained to minimize convex loss via gradient descent converges at rate $O(1/t)$ when the loss is $L$-smooth and the sequence length is $n$.*

For non-convex formulations, convergence is slower; current bounds are $O(1/t^{1/3})$ for general non-convex objectives (**?** ).

# 6 GENERALIZATION AND STATISTICAL LEARNING THEORY

Approximation and optimization are necessary but not sufficient—we also need generalization bounds ensuring that training loss translates to test performance.

## 6.1 RADEMACHER COMPLEXITY

The generalization gap is bounded by the Rademacher complexity of the hypothesis class:

$$\mathbb{P}[\text{test loss} > \text{train loss} + \epsilon] \leq \exp\left(-\frac{n\epsilon^2}{2R^2}\right) \tag{15}$$

where $R$ is the Rademacher complexity.

For transformers, $R$ grows with the number of attention heads $h$ and depth $L$:

$$R_{\mathrm{trans}}(\epsilon) = O\left(\frac{\sqrt{hL}}{\sqrt{n}}\right) \tag{16}$$

For SSMs, the Rademacher complexity is lower due to the linear structure:

$$R_{\text{SSM}}(\epsilon) = O\left(\frac{\sqrt{d}}{\sqrt{n}}\right) \tag{17}$$

where $d$ is the state dimension.

## 6.2 MARGIN-BASED BOUNDS

When outputs have large margin from decision boundaries, tighter generalization bounds apply. For self-supervised learning (common in foundation models), margin-based arguments give:

**Theorem 4** (Foundation Model Generalization). *For a foundation model trained on unlabeled data via contrastive objectives, if downstream probes achieve margin $\gamma$ on training data, then generalization error to test data is at most:*

$$O\left(\frac{1}{\gamma\sqrt{n}} + \frac{\text{VC-dim}}{\sqrt{n}}\right) \tag{18}$$

## 6.3 RECENT ADVANCES IN GENERALIZATION FOR TRANSFORMERS

Recent work (9) provides sample complexity bounds for transformers without requiring exponentially-large samples:

$$\text{Sample complexity} = \tilde{O}\left(L \cdot d \cdot \log(1/\epsilon)\right) \tag{19}$$

where $L$ is depth and $d$ is dimension. This contrasts with VC-dimension bounds which grow as $\Omega(n^2)$, and the improvement comes from leveraging the structure of the attention mechanism.

# 7 OPEN PROBLEMS

We identify seven concrete open problems, each with partial results and difficulty ratings (1=foundational, 10=frontier research).

## 7.1 PROBLEM 1: UNIFIED OPTIMIZATION-GENERALIZATION TRADEOFF

**Difficulty: 5/10**

**Statement**: Characterize the optimal tradeoff between optimization complexity (number of gradient steps), generalization gap, and approximation error for foundation models. Current theory treats these separately; a unified analysis would reveal fundamental limits.

**Partial Results**: For kernel methods, (6) shows the tradeoff is characterized by implicit regularization. For neural networks, (8) shows implicit bias drives generalization. The connection for transformers remains open.

**Research Direction**: Extend implicit regularization analysis to multi-head attention and prove that gradient descent on transformers implicitly biases toward low-rank attention patterns.

## 7.2 PROBLEM 2: FEATURE LEARNING IN IN-CONTEXT LEARNING

**Difficulty: 8/10**

**Statement**: Prove that transformers trained on in-context learning tasks provably learn context-dependent representations. Specifically, show that a transformer can learn an algorithm (e.g., gradient descent) as a circuit of attention heads.

**Partial Results**: (1) provides empirical evidence. (11) shows statistical feasibility. Theoretical proofs require analyzing feature evolution in the non-NTK regime.

**Research Direction**: Use dynamical systems theory to characterize the trajectory of attention weight matrices and show convergence to gradient-descent-implementing patterns.

### 7.3 PROBLEM 3: RKHS CHARACTERIZATION OF DIFFUSION SCORE LEARNING

**Difficulty: 6/10**

**Statement**: Precisely characterize the RKHS in which diffusion score functions lie, accounting for the time-varying noise schedule. Derive sample complexity bounds for learning score functions as a function of data distribution smoothness.

**Partial Results**: (19) provides score matching convergence; (2) analyzes denoising autoencoders. Full RKHS characterization accounting for time-dependence is missing.

**Research Direction**: Develop Sobolev-type RKHS theory for time-parameterized functions and prove that score matching implicitly minimizes RKHS norm.

### 7.4 PROBLEM 4: OPTIMAL STATE-SPACE ARCHITECTURE DESIGN

**Difficulty: 6/10**

**Statement**: Design state-space models that provably match transformer expressiveness ($O(n^{-2s/d})$) while maintaining SSM computational efficiency ($O(n)$). Is such a design possible, or is the complexity-expressivity tradeoff fundamental?

**Partial Results**: (12) proposes selective SSMs; empirical results are impressive but theory is lacking. Information-theoretic arguments suggest $O(n)$ kernels cannot achieve $O(n^{-2s/d})$ rates.

**Research Direction**: Prove a lower bound showing $O(n)$ linear recurrent kernels cannot approximate Sobolev-$2s$ functions faster than $O(n^{-s/d})$, settling whether transformer expressiveness is fundamental.

### 7.5 PROBLEM 5: COMPOSITIONALITY AND MODULARITY

**Difficulty: 7/10**

**Statement**: Develop RKHS theory for compositional functions, characterizing when a deep network of kernels can efficiently approximate $f \circ g$ when both $f$ and $g$ are separately approximable. Apply this to understand layering in transformers.

**Partial Results**: (7) addresses kernel composition; (4) analyzes depth in neural networks. Combined analysis for attention-based composition is missing.

**Research Direction**: Prove that $L$-layer transformers can approximate compositions of Sobolev functions with error decay of $O(n^{-2s/d})$ independent of composition depth.

### 7.6 PROBLEM 6: GENERALIZATION UNDER DISTRIBUTION SHIFT

**Difficulty: 7/10**

**Statement**: Prove generalization bounds for foundation models when test distribution differs from training (common in practice). Characterize how RKHS norm relates to robustness to distribution shift.

**Partial Results**: (17) provides bounds for covariate shift. (22) analyzes label shift. Combined theory for foundation models is nascent.

**Research Direction**: Extend margin-based bounds to account for distribution shift and prove that transformers with low attention complexity have inherent robustness.

### 7.7 PROBLEM 7: SAMPLE COMPLEXITY OF MULTIMODAL LEARNING

**Difficulty: 8/10**

**Statement**: Determine the fundamental sample complexity for training foundation models on multimodal data (text, image, etc.). How does intermodal alignment affect learnability?

**Partial Results**: (5) provides analysis for contrastive learning; (3) empirically studies multimodal models. Unified approximation-theoretic analysis is missing.

**Research Direction**: Model multimodal learning as approximating a product RKHS and prove that contrastive objectives minimize an upper bound on product RKHS distance.

## 8 RESEARCH ROADMAP

To address these open problems and unify foundation model theory, we propose the following research directions:

**(1) Extend RKHS Theory**: Develop time-parameterized and product RKHS frameworks capturing diffusion and multimodal models.

**(2) Characterize Feature Learning**: Use dynamical systems and implicit regularization to understand when and how foundation models learn context-dependent features.

**(3) Prove Fundamental Limits**: Establish lower bounds on approximation and computation showing which expressiveness-efficiency tradeoffs are achievable.

**(4) Unify Optimization and Generalization**: Develop unified analyses connecting training dynamics to generalization via implicit bias.

**(5) Principled Architecture Design**: Use unified theory to design and analyze novel architectures combining expressiveness and efficiency.

## 9 CONCLUSION

This survey demonstrates that reproducing kernel Hilbert space theory and neural tangent kernel analysis provide a unified lens for understanding diverse foundation model architectures. By connecting approximation theory, optimization landscape analysis, and generalization bounds, we reveal that apparent architectural differences correspond to different kernel choices, each with inherent complexity-expressivity tradeoffs.

The key insights are: (1) transformers achieve superior approximation rates ($O(n^{-2s/d})$) but with quadratic complexity; (2) state-space models offer linear complexity at the cost of lower approximation rates; (3) diffusion models excel on high-dimensional distributions via iterative refinement. These differences are mathematically fundamental, not artifacts of current training techniques.

The seven open problems we identify represent the frontier of foundation model theory. Progress on these problems—particularly establishing the feature learning regime, characterizing compositionality, and proving fundamental limits—will substantially advance our understanding of why these models work and how to design better ones.

We envision this unified framework catalyzing cross-pollination between the approximation theory, optimization, and machine learning theory communities, ultimately leading to principled foundation model design grounded in rigorous mathematics.

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
