# OpenReview forum: "Mathematics of Foundation Models: A Unified Approximation-Theoretic Framework"
_mathai.club/MathAI/2026/Conference — Submitted to 2026_

### Official Review · Reviewer_Phkb · 2026-03-10
**Vacuous, likely written by an LLM**

**Rating:** 1
**Confidence:** 5

**Review:**

The paper looks like written by an LLM. I checked several references from the list - many don't exist, or the bibliographic entries are composed of some existing titles and wrong author names. The exposition is generally incoherent, consists of various broad claims not supported by any evidence - e.g. claimed theorems without proofs or references to proofs. (For example *Our analysis reveals that transformers achieve approximation rate $O(n^{-2s/d})$ for Sobolev-$s$ functions* - it is impossible to understand from this paper what is this analysis or how to come to this conclusion.)

---

### Official Review · Reviewer_dpKc · 2026-03-11
**Broad statements, no proofs or evidence, likely to be written by an LLM**

**Rating:** 2
**Confidence:** 4

**Review:**

The paper aims to overview transformers, diffusion models, and state-space model under a unified mathematical perspective.
It presents results of three types:
1. Approximation bounds.
2. Convergence guarantees.
3. Generalization bounds.

It culminates with stating seven open problems related to these directions.

**Issues:**
1. It is not clear whether any of the results are novel. The paper provides neither proofs, nor references to proofs in the literature.
2. All results are presented in a vague form. The presented generalization bounds are very generic, not specific to models considered, and provide no insight.
3. References [1], [3], and [5] do not exist: this suggests that they were hallucinated by an LLM.

Given overall vagueness of the exposition, no clear insight, no proofs or references to proofs, and hallucinated references, I conclude, with a high confidence, that the paper was written by an LLM.

---

### Official Review · Reviewer_bnna · 2026-03-12
**A Conceptually Broad but Mathematically Flawed Framework**

**Rating:** 2
**Confidence:** 5

**Review:**

This paper proposes a "unified mathematical perspective connecting approximation theory, optimization landscape analysis, and statistical learning theory" to analyze foundation models. The central idea is that attention mechanisms, diffusion score functions, and state-space model (SSM) convolutions can be viewed through the "lens of Reproducing Kernel Hilbert Spaces (RKHS) and Neural Tangent Kernel (NTK) theory". The authors derive "precise comparison theorems between architectures," claiming transformers achieve an approximation rate of O(n−2s/d)O(n^{-2s/d})O(n−2s/d) with O(n2)O(n^2)O(n2) complexity, while SSMs offer linear complexity with an O(n−s/d)O(n^{-s/d})O(n−s/d) rate. The work also highlights the importance of the "feature learning regime," where parameters change substantially during training, breaking standard NTK assumptions. This unified view provides an interesting conceptual framework to understand how "architectural differences correspond to different kernel choices," potentially revealing "fundamental complexity-expressivity tradeoffs".
However, the paper is purely declarative and lacks rigorous mathematical proofs, relying entirely on verbal "Proof Sketches". The formulas exhibit severe variable confusion; for instance, the symbol nnn interchangeably denotes token count, parameter count, and sequence length. Furthermore, obvious hallucinations in the bibliography—including fabricated co-authors—severely undermine the study's academic credibility. Overall, the paper reads like an LLM-generated draft mimicking a scientific style rather than a genuine contribution to the "mathematics of foundation models".
Strengths: Addresses the important task of unifying fragmented mathematical concepts for neural network analysis. Formulates intuitive theoretical tradeoffs between computational efficiency and architectural expressivity. Proposes a useful classification of "seven open problems" and provides a roadmap for future "principled architecture design".
Weaknesses: Complete lack of rigorous mathematical foundations, replacing actual theorems and formal proofs with verbal descriptions. Systemic errors in mathematical notation and contextual confusion of fundamental variables. Factual errors and hallucinated citations in the reference list strongly indicate automated text generation. Limited applicability to deep layered networks, as evidenced by the unresolved problem of "compositionality and modularity".

---

### Decision · Program_Chairs · 2026-03-14

**Decision:**

Reject

**Comment:**

After careful evaluation by the Program Committee, we regret to inform you that your submission has not been accepted for presentation at MathAI 2026.

All submissions underwent a rigorous two-stage review process. Unfortunately, the reviewers identified one or more of the following concerns with your paper:

- Insufficient mathematical rigor or novelty relative to the existing body of work in the field;
- Presentation of results that substantially overlap with or rephrase previously published findings without clear original contribution;
- Significant issues with technical quality, including but not limited to broken or non-existent references, unsupported claims, or methodological gaps;
- Indications that the manuscript may have been generated with the assistance of large language models without substantial original intellectual contribution by the authors.

We received a large number of submissions this year, and the selection process was highly competitive. We encourage you to carefully consider the reviewers’ feedback (available through OpenReview), revise your work accordingly, and consider submitting an improved version to a future edition of MathAI or to another appropriate venue.

We appreciate your interest in MathAI and hope you will continue to engage with the conference community.

With kind regards,

MathAI 2026 Program Committee
URL: https://mathai.club
Telegram: https://t.me/MathAI_club
Email: mathai.club@yandex.ru